# Stillbirth among women in nine states in India: rate and risk factors in study of 886,505 women from the annual health survey

Noon Altijani,[1] Claire Carson,[2] Saswati Sanyal Choudhury,[3] Anjali Rani,[4] Umesh C Sarma,[5] Marian Knight,[2] Manisha Nair[2]

For numbered affiliations see end of article.

**Correspondence to**
Dr Manisha Nair;
manisha.nair@npeu.ox.ac.uk

## ABSTRACT

**Objectives** To assess the rate of stillbirth and associated risk factors across nine states in India.

**Design** Secondary analysis of cross-sectional data from the Indian Annual Health Survey (2010–2013).

**Setting** Nine states in India: Madhya Pradesh, Chhattisgarh, Rajasthan, Uttarakhand, Jharkhand, Odisha, Bihar, Assam and Uttar Pradesh.

**Participants** 886 505 women, aged 15–49 years.

**Main outcome measures** Stillbirth rate with 95% CI. Adjusted OR to examine the associations between stillbirth and (1) socioeconomic, behavioural and biodemographic risk factors and (2) complications in pregnancy (anaemia, eclampsia, other hypertensive disorders, antepartum and intrapartum haemorrhage, obstructed labour, breech presentation, abnormal fetal position).

**Results** The overall rate of stillbirth was 10 per 1000 total births (95% CI 9.8 to 10.3). Indicators of socioeconomic deprivation were strongly associated with an increase in stillbirth: rural residence (adjusted OR (aOR) 1.27, 95% CI 1.16 to 1.39), female illiteracy (aOR 1.43, 95% CI 1.17 to 1.74), low socioeconomic status (aOR 2.42, 95% CI 1.82 to 3.21), schedule caste background (aOR 1.11, 95% CI 1.04 to 1.19) and woman not in paid employment (aOR 1.15, 95% CI 1.07 to 1.24). Women from minority religious groups were at higher risk than the Hindu majority (Muslim (aOR 1.33, 95% CI 1.25 to 1.43); Christian (aOR 1.42, 95% CI 1.19 to 1.70)). While a few women smoked (<1%), around 9% reported chewing tobacco, which was associated with an increased odds of stillbirth (aOR 1.11, 95% CI 1.02 to 1.21). Adverse pregnancy and birth characteristics were also associated with stillbirth: antenatal care visits <4 (aOR 1.08, 95% CI 1.01 to 1.15), maternal age <25 years (aOR 1.29, 95% CI 1.21 to 1.37) and ≥35 years (aOR 1.16, 95% CI 1.04 to 1.29), multigravida (aOR 3.06, 95% CI 2.42 to 3.86), multiple pregnancy (aOR 1.77, 95% CI 1.47 to 2.15), assisted delivery (aOR 3.45, 95% CI 3.02 to 3.93), caesarean section (aOR 1.73, 95% CI 1.58 to 1.89), as were pregnancy complications (aOR 1.42, 95% CI 1.33 to 1.51).

**Conclusion** India is an emerging market economy experiencing a rapid health transition, yet these findings demonstrate the marked disparity in risk of stillbirth by women's socioeconomic status. Tobacco chewing and maternal and fetal complications were each found to be important modifiable risk factors. Targeting the 'at-risk'

### Strengths and limitations of this study

► India has the highest number of stillbirths, globally. This study identifies the characteristics of high-risk women and key risk factors in the Indian context using the largest available data set.

► The data were drawn from Indian Annual Health Survey (2010 to 2013), which sampled women from the nine states that account for 50% of the country's population, using methods designed to minimise selection bias.

► This study analyses a large sample (>800 000 women), providing sufficient statistical power to conduct a robust examination of a wide range of risk factors.

► The findings are generalisable for high burden states included in this study, but may not be generalisable to the rest of India.

► However, retrospective data collection can lead to recall bias, and there is the potential for some under-reporting of stillbirth.

population identified here, improved recording of stillbirths and the introduction of local reviews would be important steps to reduce the high burden of stillbirths in India.

## INTRODUCTION

Stillbirth is an important global health problem affecting over 7000 families every day and is associated with emotional, social and economic consequences.[1] In 2015, the stillbirth rate (SBR) was 18.4 per 1000 total births worldwide.[1] The progress in reducing stillbirth since 1990 has been slower than reductions in neonatal and under-five child mortality.[2] Currently, 98% of stillbirths occur in low-to-middle-income countries (LMICs)[1] and India has the highest number of stillbirths, with an estimated 592 100 deaths per year,[3] and a WHO estimated rate of 22 per 1000 total births. The Government of India has developed an Indian Newborn Action Plan which includes efforts to 'reduce stillbirths to <10 per 1000 births by 2030'.[4] A



modest reduction in India's SBR would translate into thousands of lives saved.

While previous studies have examined the immediate pregnancy-related risk factors for stillbirth such as infections during pregnancy and hypertensive disorders,[1 5] knowledge about distal risk factors such as socioeconomic, lifestyle related and comorbidities is limited.[1] The Indian government recognises the need to improve pregnancy care and institutional delivery among disadvantaged socioeconomic groups who have a higher risk of maternal and fetal death.[6] Since 2005, the government has made several efforts including cash assistance and dedicated services through community health workers with a stronger focus in the states with poor health and development indicators.[6] The objective of this study was to understand the current disparities in the risk of stillbirth in these 'high focus' states by examining the association between socioeconomic, biodemographic, behavioural and pregnancy-specific risk factors and stillbirth in nine states in India.

## METHODS

We conducted an analysis of secondary data from India's Annual Health Survey (AHS) (2010–2013). The survey covers nine states (Madhya Pradesh, Chhattisgarh, Rajasthan, Uttarakhand, Jharkhand, Odisha, Bihar, Assam and Uttar Pradesh) that account for about 50% of the country's population.[7] A total of 886 505 women for whom information about the outcome of their last pregnancy (live birth or stillbirth) was available and whose pregnancy lasted more than seven completed months (~28 weeks' gestation) were included. Figure 1 illustrates how the sample for this study was derived.

Based on the outcome of last pregnancy, women were divided into two groups, those with a live birth and those with a stillbirth. The outcome data are from the Women schedule (Section 1) that was implemented during the baseline round of the AHS in 2010–2011. Ever married women in the age group 15–49 years were asked about the outcome of their last pregnancy during the reference period 1 January 2007 to 31 December 2009, which was reported as either live birth, stillbirth or abortion. Information on gestational age at stillbirth or type of stillbirth (antepartum or intrapartum) was not available. The numerator 'stillbirth' and denominator 'total birth'=stillbirth+live birth) is from the same reference period of the survey data. Potential risk factors for stillbirth were grouped as: socioeconomic, behavioural and biodemographic based on a review of published literature. The description of variables included in each group is provided in table 1. A directed acyclic graph was used to construct a theoretical framework of relationships between the aforementioned risk factors and stillbirth (figure 2). In addition, in a subsample of the population (n=668 892), we examined the association of stillbirth with the following self-reported problems/complications during pregnancy: anaemia, eclampsia, other hypertensive disorders, antepartum haemorrhage, intrapartum haemorrhage, abnormal fetal position, breech presentation and obstructed labour.

## Statistical analysis

Overall and state-specific rates of stillbirth per 1000 total births with 95% CIs were calculated in the study population. The characteristics of women who had a stillbirth were compared with those who had a live birth. A univariable logistic regression analysis was performed to investigate the association between each potential risk factor and stillbirth. Maternal age was tested for linearity which showed the presence of a non-linear association with stillbirth (figure 3); it was therefore used as a categorical variable. A multivariable model was built using a stepwise forward regression, including risk factors that were statistically significantly associated with stillbirth during univariable analysis (based on a Wald p values of <0.05). The order in which the variables were added to the model, distal followed by intermediate then proximal factors, was informed by the theoretical framework (figure 2). Three variables, 'smoking tobacco', 'place of delivery' and 'timing of first ANC visit', were not statistically significantly associated with stillbirth at p<0.05 during the model building process, and were therefore dropped from the final models.

Calculated pairwise correlation coefficients did not show any significant collinearity between the risk factors. The following interactions were identified a priori and tested for significance using Wald test, which is a method of choice for survey-weighted data: asset index and gravidity, place of residence and place of delivery. We found a significant interaction between 'Gravidity' (number of pregnancies) and 'Asset index'. The final regression model was adjusted for this by fitting an interaction term. We conducted further analysis to examine the association between gravidity stratified by the quintiles of the asset index.

In a subsample of the study population with complete data on complications during pregnancy and medical comorbidities, we conducted univariable logistic regression analysis to investigate their individual association with stillbirth. Statistically significant associations were further examined using multivariable logistic regression models that adjusted for socioeconomic, behavioural and biodemographic risk factors found to be significantly associated with stillbirth. We compared the proportion of stillbirth in the subsample with the excluded group and the total sample which showed that in all three groups, the proportion of stillbirth was about 1% and live birth 99%. This suggests that the subsample for the specific pregnancy complication analysis was not a biased sample.

Missingness was investigated and data were assumed to be 'Missing at Random'. Three methods were used to address bias due to missing data: missing indicator method, complete case analysis, and multiple imputations.[8] The 'missing indicator' model in which missing data were grouped as a separate category was used as the final model. However, to maintain model stability, for variables that had <1% missing data, a separate category for 'missing' was not generated. All analyses accounted for the stratified, clustered nature of the data

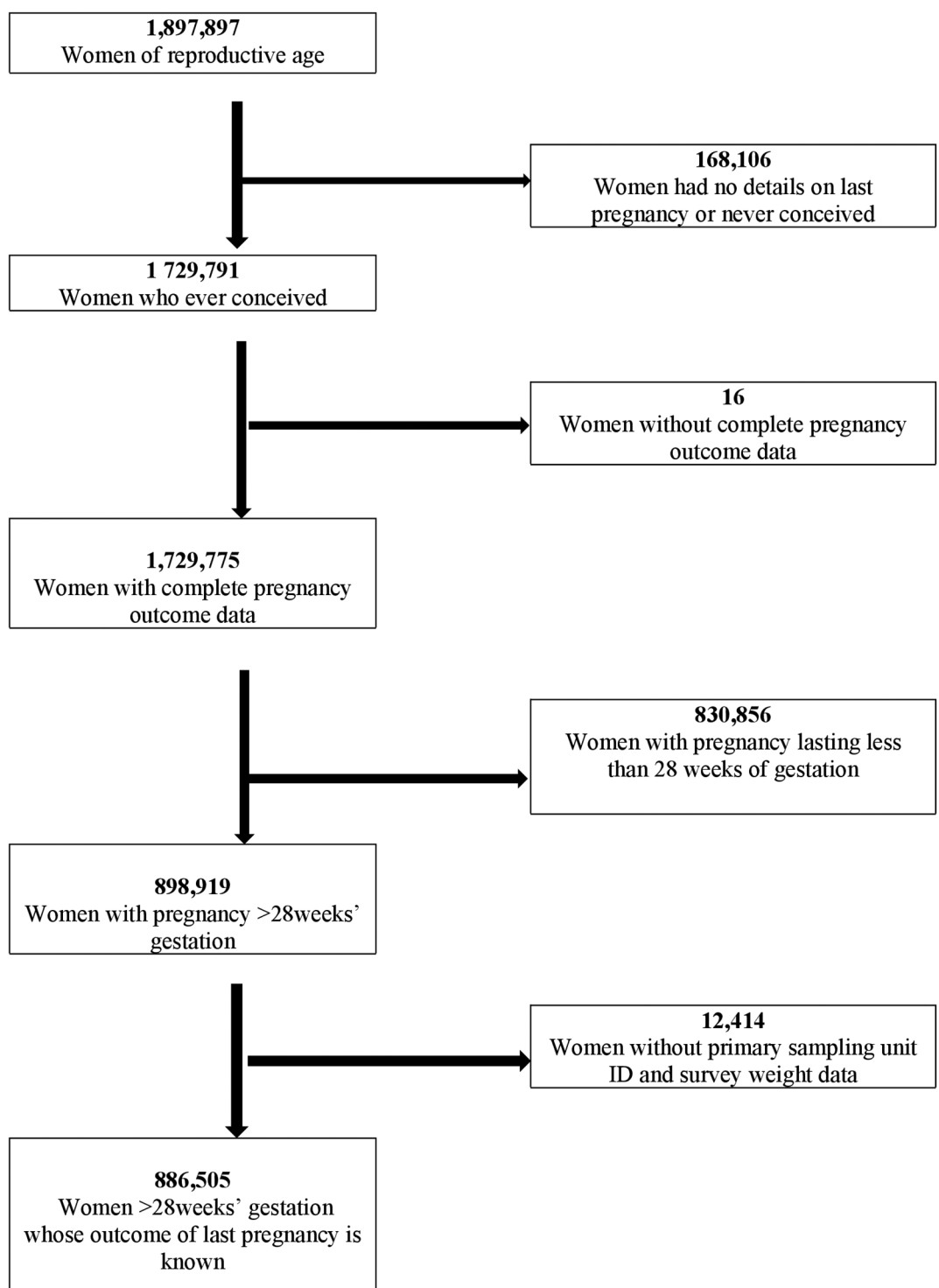

**Figure 1** Flowchart showing the derivation of the final study population.

and were conducted using the survey command (svy) in Stata V.13.1, SE (StataCorp, College Station, Texas, USA). The proportions, means and CIs presented here are weighted for design effects and non-response.

**Study power**

The fixed sample had more than 90% power to detect an OR of ≥1.11 or ≤0.91 associated with stillbirth at p<0.05 (two-tailed) for the risk factor with

the highest prevalence (83% for rural residence) and an OR of ≥1.39 or ≤0.67 for the risk factor with the lowest prevalence in the study population (current smoker 0.8%).

**Patient and public involvement**

This is not applicable since this was a secondary analysis of anonymous survey data.

**Table 1**  Description of the variables used to examine the risk factors for stillbirth

| Risk factors | Description of the variables |
|---|---|
| **Proximal or biodemographic factors** | |
| **Fetal factors** | |
| Sex | Sex of the fetus was coded as either female or male. |
| Multiple pregnancy | Number of fetuses: Women were categorised as singleton (one fetus) or multiple pregnancy (twins or higher-order multiple gestations). |
| **Maternal factors** | |
| Maternal age | Maternal age at the time of the survey was divided into 5-year age bands: <20, 20–24, 25–29, 30–34, 35–39, 40–44, 45 and above. |
| Gravidity | Gravidity (number of pregnancies) was used as a proxy for parity (number of deliveries) for which information was not available. Gravidity was categorised as 'first' if the index pregnancy was the first pregnancy, 'second - fourth' and 'five or more'. |
| Any complication during pregnancy | Women who reported to have any of the following complications during their index pregnancy were coded as 'yes', otherwise 'no': anaemia, eclampsia, other hypertensive disorders, antepartum haemorrhage, intrapartum haemorrhage, abnormal fetal position, breech presentation and obstructed labour. |
| Mode of delivery | This was categorised into vaginal delivery, assisted vaginal delivery and caesarean section. |
| **Intermediate factors or behavioural factors** | |
| **Lifestyle factors** | |
| Smoking tobacco  Chewing tobacco | Women were asked about smoking and chewing tobacco during the interview; current practice was coded as 'yes', while women who never practised or no longer practised were coded as 'no'. |
| **Health seeking behaviour** | |
| Antenatal care visits | Number of antenatal care (ANC) visits was categorised as '≥4 visits' (which is recommended as adequate by the WHO), and '<4 visits'. |
| Timing of first ANC visit | Timing of the first ANC visit was categorised as 'visit in the first trimester (or first 3 months), which is recommended as adequate by the WHO, and 'after first 3 months'. |
| Place of delivery | Place of delivery was grouped as 'medical facility' or 'home'. |
| **Distal or socioeconomic factors** | |
| **Social factors** | |
| Religion | Women were categorised into the following religious groups: 'Hindu', 'Muslim', 'Christian' and 'Others' which included several groups with small numbers such as Sikh, Buddhists, etc. |
| Place of residence | Place of residence was grouped into urban and rural. |
| Social group | Women were categorised into the following social groups: 'Schedule caste (SC)', 'Schedule tribe (ST)' and 'Others'  ▶ SC and ST are officially designated groups of historically disadvantaged populations in India  ▶ 'Others' included the general social class and other backward classes |
| **Economic factors** | |
| Education | Maternal education at the time of the survey was categorised into: illiterate, primary school or below, secondary school, tertiary and above. |
| Occupation | Based on occupational status, women were grouped as 'being in paid employment' or 'not in paid employment'. |
| Economic status | Asset index was used to measure the economic status of the participants. Asset index scores were calculated and the study sample was divided into quintiles ranging from the lowest (quintile 1) to highest (quintile 5) socioeconomic status. |
| **Pregnancy complication** | |
| Anaemia during pregnancy | This is a derived variable based on questions related to the following signs and symptoms: paleness, giddiness, weakness, excessive fatigue. |

## RESULTS

### Rate of stillbirth

Of the 886 505 women included in the analyses, 8429 reported a stillbirth, giving an overall rate of 10 stillbirths (95% CI 9.8 to 10.3) per 1000 total births. The rates of stillbirth per state are shown in table 2 and figure 4.

### Association between socioeconomic, behavioural and biodemographic characteristics and stillbirth

The median age for women in the study population was 26 years (IQR 23–30). The frequency, proportions and association with stillbirth (unadjusted and adjusted OR (aOR)) for socioeconomic, behavioural and biodemographic

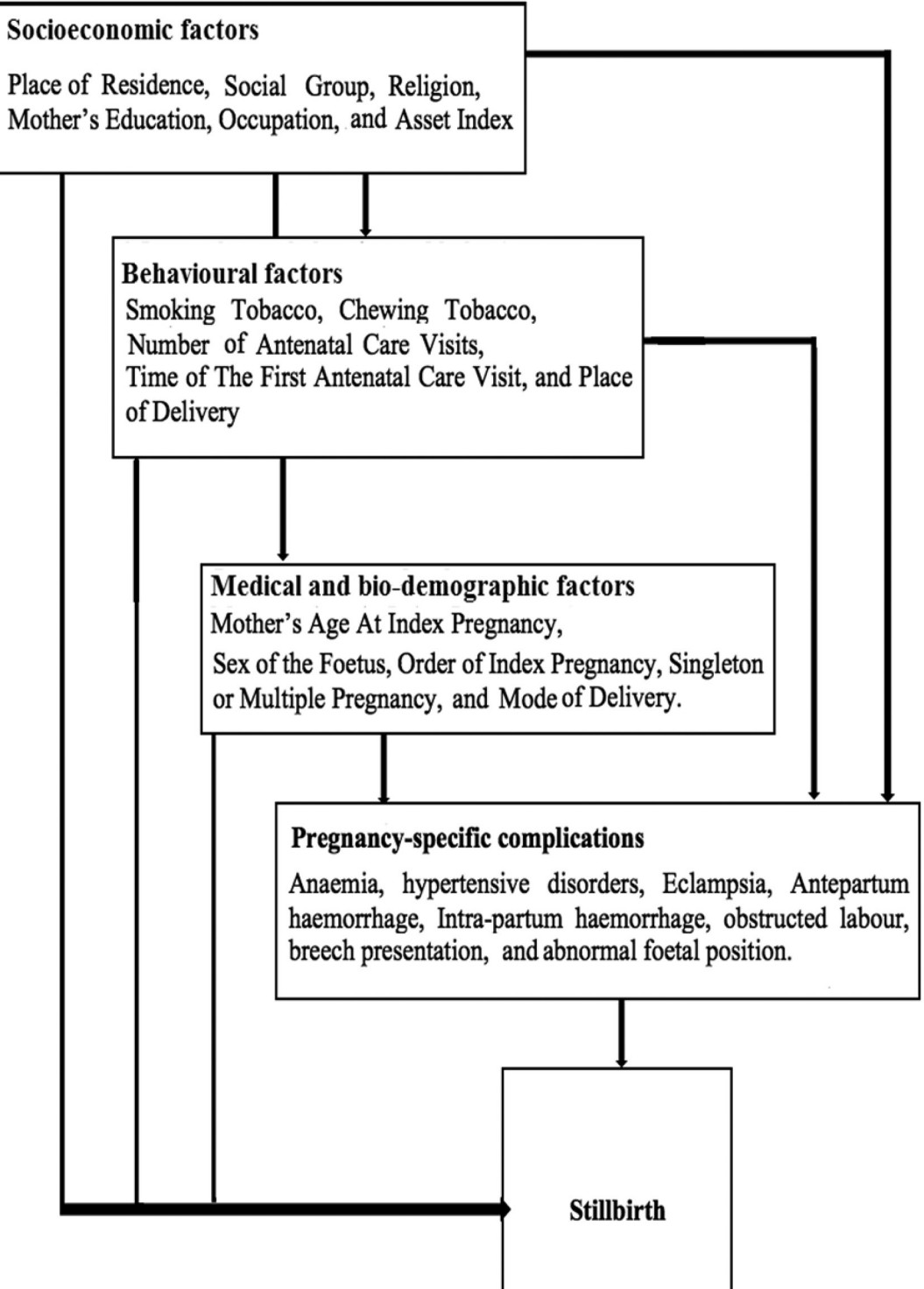

**Figure 2** Hypothesised relationship between the risk factors and stillbirth.

variables are shown in table 3, and pregnancy-related variables in table 4. After adjusting for other risk factors, all examined socioeconomic factors were found to be significantly associated with stillbirth. Women living in rural settings had 27% higher odds of stillbirth compared with urban settings. Belonging to a schedule caste social group was associated with 11% higher odds compared with the reference 'other' social group, but 26% lower odds of stillbirth for women belonging to the schedule tribe

group. Compared with Hindus, belonging to Muslim and Christian religious groups was associated with increased odds of stillbirth. Compared with women with university education or higher, women with no school education had 43% higher odds. Women not in paid employment were 15% more likely to have a stillborn baby compared with women in paid employment.

Chewing tobacco was associated with 11% higher odds of stillbirth after adjusting for other risk factors in the

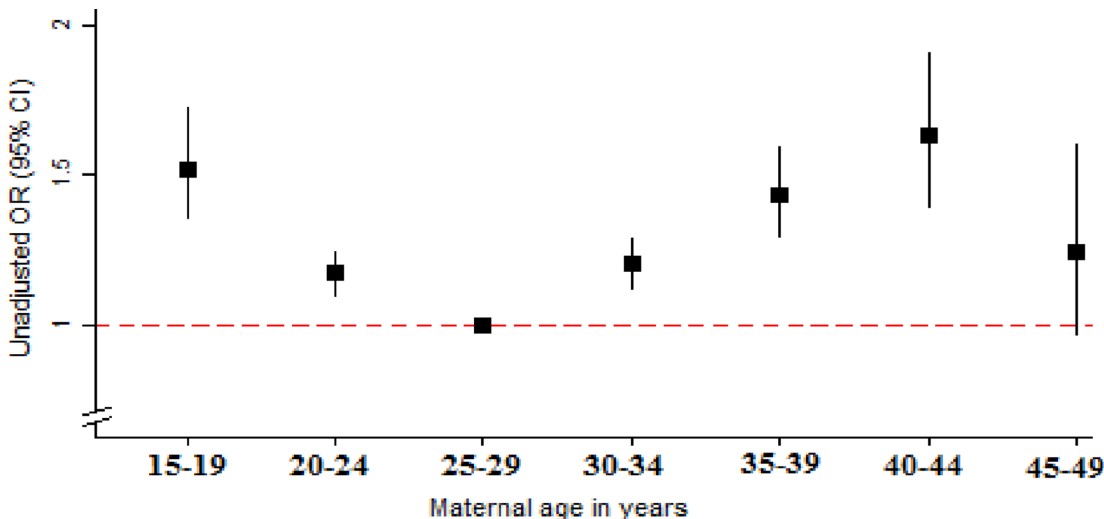

**Figure 3** Association between maternal age in 5-year groups and stillbirth. Data source: Annual Health Survey 2012–2013, India.

study population. Health seeking behaviours were significantly associated with stillbirth. Women attending <4 antenatal care (ANC) visits had 8% higher odds compared with women who had ≥4 ANC visits.

All examined demographic and pregnancy-related factors were also found to be significantly associated with stillbirth in the study population. After full adjustment, the non-linear (U-shaped) association between mother's age and stillbirth persisted. Multiple pregnancies were associated with 77% higher odds of stillbirth compared with singleton. Pregnancies with a male fetus had 26% higher odds compared with a female fetus. Women who had an assisted delivery or caesarean section were more likely to report a stillbirth compared with women who had a normal delivery.

We found a significant statistical interaction between 'gravidity' and 'Asset index'. A stratified analysis showed that multigravida women had higher odds of stillbirth and this association was substantially stronger for women who

belonged to the higher asset index quintiles (table 5). The results of sensitivity analyses (complete case analysis and multiple imputations) were not materially different from the 'missing indicator' model (see online supplementary table S1).

### Association between specific complications during pregnancy and stillbirth

Reporting any pregnancy complication was associated with significantly higher odds of stillbirth after adjusting for the identified socioeconomic, behavioural and biodemographic risk factors (table 4). The association between each of the reported complication and stillbirth are shown in table 6. Women who reported to have anaemia during pregnancy had 35% higher odds of stillbirth compared with women who did not have anaemia. Eclampsia was associated with almost twice the odds of stillbirth and other hypertensive disorders were associated with about 22% higher odds. Women who had an antepartum or

**Table 2** Number and rate of stillbirths in the study population in nine states in India

| State | Stillbirths | Total births | Stillbirth rate per 1000 total birth (95% CI) |
|---|---|---|---|
| Assam | 856 | 64 841 | 12.8 (11.8 to 13.9) |
| Bihar | 2833 | 250 609 | 11.3 (10.9 to 11.8) |
| Chhattisgarh | 476 | 98 220 | 5.0 (4.5 to 5.5) |
| Jharkhand | 657 | 60 153 | 10.6 (9.7 to 11.6) |
| Madhya Pradesh | 615 | 145 552 | 4.2 (3.8 to 4.6) |
| Odisha | 534 | 46 162 | 10.9 (9.9 to 12.0) |
| Rajasthan | 413 | 52 537 | 7.1 (6.4 to 8.0) |
| Uttar Pradesh | 1813 | 131 324 | 14.8 (14.1 to 15.6) |
| Uttarakhand | 233 | 37 485 | 7.7 (6.5 to 9.2) |
| Overall | 8430 | 886 505 | 10.0 (9.8 to 10.3) |

Frequencies are unweighted (true counts). Rates are weighted for design effects and non-response.

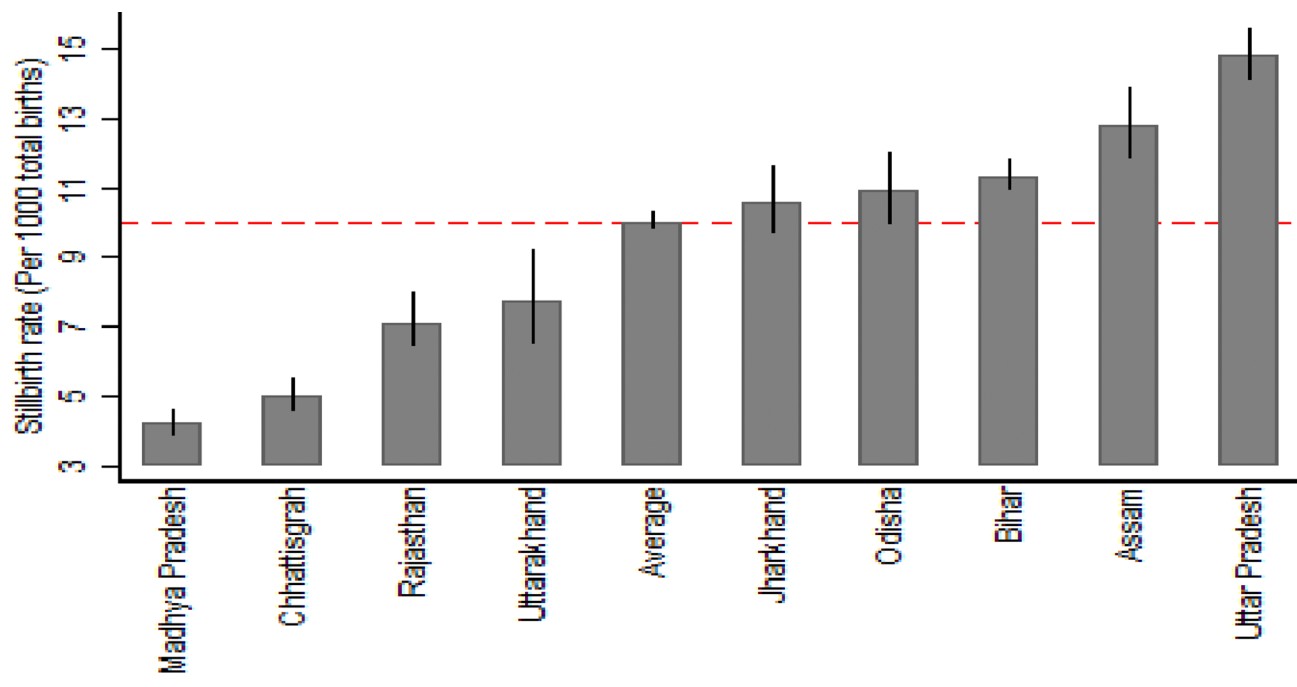

**States in India**

**Figure 4** Rate of stillbirth overall and by nine states in India using the Annual Health Survey data (2010–2013).

intrapartum haemorrhage had significantly higher odds of stillbirth compared with women who did not have these complications. The adjusted odds of stillbirth were 58% higher in women with abnormal fetal position, more than three times higher for women who had an obstructed labour and nearly three times higher for women who had a breech presentation compared with women who did not report these complications. We also noted that the association between stillbirth and caesarean section was no longer significant at p<0.05 and the aOR for assisted delivery was significantly attenuated in the model that examined the association between obstructed labour and stillbirth. Sensitivity analysis conducted using complete case analysis and multiple imputations did not change the results materially (see online supplementary table S2).

### DISCUSSION

The overall rate of stillbirth was found to be 10 per 1000 total births, but varied widely between the nine states ranging from 4.2 to 14.8 per 1000 total births. The marked variation at the state level may be explained by the different population characteristics and the varying distribution of risk factors across the nine states. The findings show the impact of inequality on stillbirth, as women in the most deprived groups were at highest risk. This was identified across a number of variables which capture different aspects of deprivation; for example, poorer women, those with little education, those living in rural areas, not currently working and belonging to 'schedule caste' groups were at increased risk of stillbirth compared with more affluent and advantaged women. Yet, among

the multigravidas, odds of stillbirth were substantially higher in the more affluent group, after adjusting for other factors. Chewing tobacco is highlighted as a significant risk factor for stillbirth, which is important in the Indian context where few women report tobacco smoking. Pregnancy complications, including anaemia, eclampsia, other hypertensive disorders, antepartum haemorrhage, intrapartum haemorrhage, abnormal fetal position, breech presentation and obstructed labour significantly increased the odds of stillbirth.

The overall estimated SBR in the study population from the nine states in India was approximately half that of the WHO estimated rate of 22 per 1000 total births. This was also true for the rate of stillbirth for the state of Bihar (11.3 per 1000 births; 95% CI 10.9 to 11.8 reported in table 2) when compared with the rate estimated for this state by another study that used both household survey and verbal autopsy to estimate the rate (21.2 per 1000 births; 95% CI 19.7 to 22.6).[9] However, the risk factors for stillbirth identified by our study is consistent with the findings of the other study by Dandona *et al.*[9] Possible explanations for the low SBR include under-reporting of stillbirths or classifying stillbirths as neonatal deaths due to associated stigma. It is also important to acknowledge that the rates of stillbirth from the WHO, from Dandona *et al*'s study[9] and from our study are all estimates, and it is difficult to confirm which estimate is the most accurate.

We found a significant disparity in the risk of stillbirth by socioeconomic status which is consistent with previous research. A systematic review of studies from developing countries showed that low socioeconomic status was

**Table 3** The association between stillbirth and socioeconomic behavioural and biodemographic factors: frequencies, unadjusted and adjusted ORs

| Variables | Total births N=886 505 Frequency (%) | Stillbirths N=8429 Frequency (%) | Live births N=878 076 Frequency (%) | Unadjusted OR (95% CI*) | Adjusted OR† (95% CI*) |
|---|---|---|---|---|---|
| Maternal age (years) | | | | | |
| 15–19 | 33 482 (3.7) | 436 (4.9) | 33 046 (3.7) | 1.52 (135 to 1.72) | 1.76 (1.55 to 2.00) |
| 20–24 | 324 130 (36.0) | 3079 (36.2) | 321 051 (36.0) | 1.17 (1.09 to 1.24) | 1.29 (1.21 to 1.37) |
| 25–29 | 301 330 (33.7) | 2546 (29.2) | 298 784 (33.7) | 1.00 (Ref) | 1.00 (Ref) |
| 30–34 | 143 529 (16.6) | 1400 (17.1) | 142 129 (16.6) | 1.20 (1.11 to 1.29) | 1.05 (0.97 to 1.14) |
| 35–39 | 55 893 (6.6) | 643 (8.2) | 55 250 (6.6) | 1.43 (1.29 to 1.59) | 1.16 (1.04 to 1.29) |
| 40–45 | 20 006 (2.4) | 244 (3.4) | 19 762 (2.4) | 1.63 (1.39 to 1.91) | 1.29 (1.09 to 1.51) |
| 45–49 | 8135 (1.0) | 81 (1.1) | 8054 (1.0) | 1.24 (0.96 to 1.60) | 1.04 (0.80 to 1.35) |
| Place of residence | | | | | |
| Urban | 123 099 (16.4) | 874 (12.7) | 122 225 (16.4) | 1.00 (Ref) | 1.00 (Ref) |
| Rural | 763 406 (83.7) | 7555 (87.4) | 755 851 (83.6) | 1.35 (1.24 to 1.47) | 1.27 (1.16 to 1.39) |
| Religion | | | | | |
| Hindu | 734 134 (81.5) | 6522 (75.3) | 727 612 (81.5) | 1.00 (Ref) | 1.00 (Ref) |
| Muslim | 125 723 (15.9) | 1619 (21.6) | 124 104 (15.8) | 1.48 (1.39 to 1.58) | 1.33 (1.25 to 1.43) |
| Christian | 14 914 (1.8) | 178 (2.4) | 14 736 (1.8) | 1.49 (1.25 to 1.78) | 1.42 (1.19 to 1.70) |
| Others | 11 734 (0.9) | 110 (0.7) | 11 624 (0.9) | 0.87 (0.69 to 1.11) | 1.06 (0.84 to 1.35) |
| Social group | | | | | |
| Other | 598 477 (68.4) | 5868 (70.0) | 592 609 (68.4) | 1.00 (Ref) | 1.00 (Ref) |
| Schedule caste | 161 273 (19.2) | 1644 (21.4) | 159 629 (19.2) | 1.09 (1.02 to 1.16) | 1.11 (1.04 to 1.19) |
| Schedule tribe | 126 755 (12.4) | 917 (8.7) | 125 838 (12.5) | 0.68 (0.62 to 0.74) | 0.74 (0.68 to 0.81) |
| Maternal education | | | | | |
| Tertiary and above | 36 617 (3.9) | 188 (2.2) | 36 429 (4.0) | 1.00 (Ref) | 1.00 (Ref) |
| Secondary | 110 458 (11.3) | 786 (8.2) | 109 672 (11.3) | 1.31 (1.08 to 1.60) | 1.10 (0.90 to 1.34) |
| Primary and below | 364 809 (39.1) | 3413 (38.1) | 361 396 (39.1) | 1.77 (1.48 to 2.12) | 1.34 (1.10 to 1.62) |
| Illiterate | 374 621 (45.7) | 4042 (51.6) | 370 579 (45.6) | 2.06 (1.72 to 2.47) | 1.43 (1.17 to 1.74) |
| Employment | | | | | |
| In paid employment | 154 177 (16.4) | 1273 (13.2) | 152 904 (16.4) | 1.00 (Ref) | 1.00 (Ref) |
| Not in-paid employment | 732 328 (83.6) | 7156 (86.8) | 725 172 (83.6) | 1.29 (1.20 to 1.38) | 1.15 (1.07 to 1.24) |
| Asset index, quintiles | | | | | |
| 5='Highest' | 159 957 (17.9) | 1133 (13.6) | 158 824 (17.9) | 1.00 (Ref) | 1.00 (Ref) |
| 4 | 160 335 (17.6) | 1422 (16.9) | 158 913 (17.6) | 1.26 (1.14-1.38) | 1.34 (1.01 to 1.79) |
| 3 | 159 567 (18.0) | 1606 (19.3) | 157 961 (18.0) | 1.41 (1.29 to 1.55) | 1.91 (1.45 to 2.52) |
| 2 | 157 506 (17.9) | 1654 (19.8) | 155 852 (17.8) | 1.46 (1.33 to 1.60) | 2.45 (1.85 to 3.24) |
| 1='Lowest' | 155 283 (18.0) | 1626 (18.9) | 153 657 (17.9) | 1.39 (1.26 to 1.52) | 2.42 (1.82 to 3.21) |
| Missing | 93 857 (10.7) | 988 (11.5) | 92 869 (10.7) | 1.41 (1.27 to 1.56) | 1.21 (0.91 to 1.62) |
| Current smoking‡ | | | | | |
| No | 725 560 (82.0) | 6670 (79.6) | 718 890 (82.0) | 1.00 (Ref) | – |
| Yes | 6553 (0.8) | 80 (1.2) | 6473 (0.8) | 1.53 (1.19 to 1.97) | – |

**Table 3**  Continued

| Variables | Total births N=886 505 Frequency (%) | Stillbirths N=8429 Frequency (%) | Live births N=878 076 Frequency (%) | Unadjusted OR (95% CI*) | Adjusted OR† (95% CI*) |
|---|---|---|---|---|---|
| Missing | 154 392 (17.2) | 1679(19.3) | 152 713 (17.2) | 1.16 (1.08 to 1.23) | – |
| Chewing tobacco‡ | | | | | |
| No | 635 393 (74.1) | 5743 (71.5) | 629 650 (74.1) | 1.00 (Ref) | 1.00 (Ref) |
| Yes | 96 806 (8.7) | 1010 (9.3) | 95 796 (8.7) | 1.10 (1.01 to 1.19) | 1.11 (1.02 to 1.21) |
| Missing | 154 306 (17.2) | 1676 (19.2) | 152 630 (17.2) | 1.16 (1.09 to 1.24) | 1.44 (1.29 to 1.59) |

Frequencies are true counts, % and OR are weighted for design effects and non-response.

*The 95 % CIs were calculated using linearised SEs.

†Multivariable logistic regression, adjusting for the other variables in the model, except self-reported mental illness. The model also adjusts for the observed significant interaction between gravidity and asset index. Smoking was not significantly associated with stillbirth during the model-building process and therefore removed from the final model.

‡These variables reflect practice at the time of the survey.

significantly associated with stillbirth with a population attributable fraction ranging between 2% and 75%.[10] A study from 13 European countries reported that lower maternal education and maternal unemployment were associated with 1.9 and 1.6 times higher odds of stillbirth, respectively.[10] Maternal education and employment may act through promoting high self-esteem and empowering women to make decisions about healthcare utilisation.[11]

The increased odds of stillbirth in women belonging to religious minority groups in India could be due to cultural constraints reported in other studies or inequalities that minority groups are subjected to which influences their healthcare seeking behaviours, even in countries that have universal access to medical care.[12] In India, schedule tribe and schedule caste social groups have been historically disadvantaged; belonging to a schedule caste group was associated with higher odds of stillbirth which is consistent with previous literature of disadvantaged groups having higher risk.[13] However, women belonging to the schedule tribe group had lower odds of stillbirth. While, it might be possible that individuals belonging to schedule tribe groups have different health seeking behaviours leading to a lower risk,[14 15] other factors, not adjusted for directly in this study, could explain this difference. Examples include nutritional risk factors, exposure to environmental toxin through occupation or household (such as indoor air pollution) or exposure to domestic violence.

Self-reported tobacco smoking or chewing has been shown to be associated with increased risk of stillbirth in other studies.[16 17] In the study population, chewing tobacco was 15 times more common compared with smoking tobacco and women who chewed tobacco had a 14% higher odds of stillbirth compared with women who did not. In contrast to other studies,[16 17] smoking was not found to be significantly associated with stillbirth in our study population after adjusting for other risk factors (aOR 1.26; 95% CI 0.97 to 1.63). This could be because smoking at the time of the survey may not reflect smoking during pregnancy. While smoking during pregnancy is

discouraged, healthcare messages about chewing tobacco during pregnancy are less clear.

Our findings related to timing and number of ANC visits conforms to the results of other studies. A study by Chopra[18] et al estimated that 24% of stillbirth and perinatal deaths in South Africa could be prevented every year through improved use of ANC services. However, another study from South Africa found that timing of booking visit may not, in isolation, be an important determinant of stillbirth.[19] Improving the quality of pregnancy care with specific measures to prevent stillbirth are important in addition to increasing coverage of antenatal care. Assisted delivery and caesarean section were found to be associated with higher risk of stillbirth, but this association was grossly attenuated and no longer statistically significant for caesarean section after adjusting for obstructed labour, suggesting the presence of a reverse causation. Similar to other studies, our study showed a U-shaped association between maternal age and stillbirth[20–22] and male sex of the fetus to be associated with higher odds of stillbirth.[1 23] The higher odds of stillbirth observed with multiple pregnancies compared with singleton pregnancies can be explained by the propensity of women with multiple pregnancies to have more complications.[24]

A number of studies have shown that anaemia during pregnancy is associated with increased risk of stillbirth, with 3.7–16 times higher odds of stillbirth associated with anaemia among pregnant women.[25 26] A meta-analysis of risk factors in high-income countries reported higher odds of stillbirth for women with pregnancy-induced hypertension, pre-eclampsia and eclampsia.[27] Antepartum and intrapartum haemorrhage can be due to a myriad of obstetric complications; a meta-analysis reported that there was a four-fold increased risk of stillbirth associated with bleeding.[28] Breech presentation, abnormal fetal position and obstructed labour operate through similar mechanisms; the fetus is trapped in the birth canal and is subjected to hypoxia leading to stillbirth.[29] However, a stillbirth could be prevented if these

**Table 4** The association between stillbirth and pregnancy-related factors: frequencies, unadjusted and adjusted ORs

| Variables | Total births N=886 505 Frequency (%) | Stillbirths N=8429 Frequency (%) | Live births N=878 076 Frequency (%) | Unadjusted OR (95% CI*) | Adjusted OR† (95% CI*) |
|---|---|---|---|---|---|
| **Number of antenatal care visit** | | | | | |
| Four or more | 269 148 (28.9) | 2222 (24.6) | 266 926 (29.0) | 1.00 (Ref) | 1.00 (Ref) |
| Less than four | 518 288 (58.7) | 4939 (59.3) | 513 349 (58.6) | 1.19 (1 12 to 1.26) | 1.08 (1.01 to 1.15) |
| Missing | 99 069 (12.4) | 1268 (16.1) | 97 801 (12.4) | 1.53 (1.41 to 1.66) | 1.36 (1.25 to 1.48) |
| **Timing of the first antenatal care visit** | | | | | |
| Three or less | 534 118 (58.6) | 4678 (54.0) | 529 440 (58.7) | 1.00 (Ref) | – |
| More than three | 253 460 (28.9) | 2483 (29.9) | 250 977 (28.9) | 1.12 (1.06 to 1.19) | – |
| Missing | 98 927 (12.4) | 1268 (16.1) | 97 659 (12.4) | 1.41 (1.31 to 1.52) | – |
| **Place of delivery** | | | | | |
| Medical facility | 539 704 (60.5) | 5176 (61.9) | 534 528 (60.5) | 1.00 (Ref) | – |
| Home | 346 801 (39.5) | 3253 (38.1) | 343 548 (39.5) | 0.94 (0.89 to 0.99) | – |
| **Number of pregnancies** | | | | | |
| One | 158 886 (16.9) | 957 (10.6) | 157 929 (16.9) | 1.00 (Ref) | 1.00 (Ref) |
| Two to four | 414 938 (46.8) | 4409 (51.8) | 410 529 (46.7) | 1.77 (1.63 to 1.93) | 3.06 (2.42 to 3.86) |
| Five or more | 93 458 (11.7) | 1404 (19.0) | 92 024 (11.7) | 2.60 (2.36 to 2.87) | 4.98 (3.66 to 6.74) |
| Missing | 219 253 (24.7) | 1659 (18.6) | 217 594 (24.7) | 1.21 (1.10 to 1.32) | 1.40 (0.68 to 2.90) |
| **Sex of the baby** | | | | | |
| Female | 414 913 (46.6) | 3468 (41.2) | 411 445 (46.7) | 1.00 (Ref) | 1.00 (Ref) |
| Male | 471 592 (53.4) | 4961 (58.9) | 466 631 (53.3) | 1.25 (1.19 to 1.32) | 1.26 (1.20 to 1.33) |
| **Multiple gestations (index pregnancy)** | | | | | |
| Singleton | 878 556 (99.1) | 8267 (98.2) | 870 289 (99.1) | 1.00 (Ref) | 1.00 (Ref) |
| Twin pregnancy | 7949 (0.9) | 162 (1.8) | 7787 (0.9) | 2.03 (1.67 to 2.45) | 1.77 (1.47 to 2.15) |
| **Mode of delivery** | | | | | |
| Spontaneous vaginal | 805 430 (90.7) | 7146 (84.3) | 798 284 (90.8) | 1.00 (Ref) | 1.00 (Ref) |
| Assisted vaginal | 14 909 (1.8) | 390 (5.0) | 14 519 (1.7) | 3.13 (2.76 to 3.55) | 3.45 (3.02 to 3.93) |
| Caesarean section | 66 166 (7.6) | 893 (10.7) | 65 273 (7.5) | 1.53 (1.40 to 1.66) | 1.73 (1.58 to 1.89) |
| **Any pregnancy complications** | | | | | |
| No | 250 099 (28.1) | 2000 (23.5) | 250 099 (28.1) | 1.00 (Ref) | 1.00 (Ref) |
| Yes | 418 856 (47.5) | 4779 (58.0) | 414 077 (47.4) | 1.47 (1.38 to 1.56) | 1.42 (1.33 to 1.51) |
| Missing | 217 550 (24.5) | 1650 (18.5) | 215 900 (24.5) | 0.91 (0.84 to 0.98) | 1.99 (1.00 to 3.94) |

Frequencies are true counts, % and OR are weighted for design effects and non-response.
*Multivariable logistic regression, adjusting for the other variables in the model, except self-reported mental illness. The model also adjusts for the observed significant interaction between gravidity and asset index.
†The 95% CIs were calculated using linearised SEs.

complications are detected timely and managed appropriately. Previous studies suggest that women with pre-existing mental health problems have 70% higher risk of stillbirth.[30]

The AHS is the largest health survey in the world. The large sample provided adequate number of events (stillbirths) and allowed us to conduct an adequately powered, robust examination of a wide range of risk factors. The survey is a representative sample from nine states in India with high rates of stillbirth and the sampling strategy minimised the possibility of selection bias. The findings are generalisable for high burden states

**Table 5** Adjusted* ORs and 95% CI for the association between gravidity and stillbirth stratified by the asset index quintiles

**A. Missing indicator method**

| Asset index quintiles | | | | | |
|---|---|---|---|---|---|
|  | 1 (most deprived) | 2 | 3 | 4 | 5 (least deprived) |
| Gravidity | | | | | |
| 1 | 1.00 (Ref) | 1.00 (Ref) | 1.00 (Ref) | 1.00 (Ref) | 1.00 (Ref) |
| 4-Feb | 1.44 (1 15 to 1.81) | 1.47 (1.18 to 1.84) | 1.83 (1.48 to 2.26) | 2.55 (2.02 to 3.23) | 3.04 (2.37 to 3.90) |
| More than 5 | 1.89 (1.43 to 2.49) | 1.88 (1.43 to 2.48) | 3.04 (2.34 to 3.94) | 3.71 (2.75 to 5.02) | 4.62 (3.27 to 6.54) |
| Missing | 0.95 (0.74 to 1.22) | 1.18 (0.92 to 1.50) | 1.37 (1.09 to 1.72) | 1.88 (1.44 to 2.45) | 2.26 (1.67 to 3.05) |

**B. Complete case analysis**

| Asset index quintiles | | | | | |
|---|---|---|---|---|---|
|  | 1 | 2 | 3 | 4 | 5 |
| Gravidity | | | | | |
| 1 | 1.00 (Ref) | 1.00 (Ref) | 1.00 (Ref) | 1.00 (Ref) | 1.00 (Ref) |
| 4-Feb | 1.19 (0.92 to 1.54) | 1.22 (0.94 to 1.60) | 1.74 (1.35 to 2.24) | 2.05 (1.56 to 2.67) | 2.86 (2.11 to 3.88) |
| More than 5 | 1.59 (1.15 to 2.18) | 1.48 (1.05 to 2.08) | 2.70 (1.97 to 3.73) | 3.02 (2.12 to 4.31) | 5.02 (3.35 to 7.53) |

**C. Multiple imputations**

| Asset index quintiles | | | | | |
|---|---|---|---|---|---|
|  | 1 | 2 | 3 | 4 | 5 |
| Gravidity | | | | | |
| 1 | 1.00 (Ref) | 1.00 (Ref) | 1.00 (Ref) | 1.00 (Ref) | 1.00 (Ref) |
| 4-Feb | 1.48 (0.90 to 2.48) | 1.50 (0.90 to 2.55) | 1.77 (1.08 to 2.92) | 2.29 (1.39 to 3.84) | 2.73 (2.20 to 3.40) |
| More than 5 | 1.90 (1.00 to 3.52) | 1.98 (1.07 to 3.81) | 2.69 (1.40 to 5.04) | 3.31 (1.70 to 6.45) | 4.41 (3.33 to 5.86) |

Results are weighted for design effects and non-response.
*Multivariable logistic regression, adjusting for socioeconomic, health seeking behaviour, pre-existing medical conditions and biodemographic characteristics.

included in this study, but may not be generalisable to the rest of India. To our knowledge, no other large data set is available to investigate stillbirth in India, the country with the highest number of stillbirths.

There are some limitations of this study which should be considered when interpreting the findings. Important to consider is that stillbirths may be under-reported or misclassified in the AHS from which our data were drawn. A study by Christou et al[31] showed that stillbirths are likely to be under-reported in routine household surveys, but this was more likely to be due to a lack of rigorous ascertainment of pregnancy outcomes rather than deliberate non-reporting by women due to any reason. We did not find any evidence from published literature suggesting under-reporting of stillbirth to vary by risk factors or specific population groups; thus, we excluded the possibility of differential under-reporting. However, we cannot exclude the possibility of misclassification of stillbirth as miscarriage/abortion or neonatal deaths as stillbirth. A small proportion of the stillbirths were reported as pregnancy loss/abortion after 7 months of gestation in the data set. We reclassified these as stillbirth in our analysis.

We acknowledge that household surveys are not the ideal source of data for stillbirth, but at present, they are the only source of data for a majority of the countries.

The reliability of the reporting of stillbirth in household surveys could be improved by including information on gestational age at stillbirth, probing questions as was done in a study by Dandona et al conducted in one state in India[9] or by using verbal autopsy[32]; but currently, it is not being used in large household surveys due to cost implications.

Retrospective data collection could also have led to potential recall bias, and self-reported complications during pregnancy were not verified with hospital records. It was not possible to differentiate between pregnancy-induced hypertension and pre-existing essential hypertension. To minimise the effect of recall bias, dummy variables (eg, anaemia) were generated wherever feasible and highly subjective symptoms (eg, prolonged labour) were not investigated. Data were not available for a number of potential risk factors for stillbirth such as timing of stillbirth, thus we were not able to analyse the risk factors separately for antepartum and intrapartum stillbirths. An inherent limitation of routine survey data is missing information; however, the results of the analyses using three different methods to account for missing data were not substantially different.

## CONCLUSION

Our study showed that despite several efforts being made to improve pregnancy care in India, socioeconomic

**Table 6** Association between stillbirth and complications during pregnancy—frequencies, unadjusted and adjusted ORs (95% CI)

| Variables | Total births N=668 892 Frequency (%) | Stillbirths N=6777 Frequency (%) | Live births N=662 115 Frequency (%) | Unadjusted ORs (95% CI) | Adjusted ORs (95% CI)*† |
|---|---|---|---|---|---|
| Anaemia | | | | | |
| No | 500 785 (73.8) | 4611 (66.0) | 496 174 (73.9) | 1 (Ref) | 1 (Ref) |
| Yes | 168 107 (26.2) | 2166 (34.0) | 165 941 (26.1) | 1.45 (1 36 to 1.54) | 1.35 (1.27 to 1.43) |
| Eclampsia | | | | | |
| No | 628 681 (93.8) | 6043 (88.9) | 622 638 (93.8) | 1 (Ref) | 1 (Ref) |
| Yes | 40 211 (6.2) | 734 (11.2) | 39 477 (6.2) | 1.90 (1.73 to 2.10) | 1.79 (1.62 to 1.97) |
| Other hypertensive disorders | | | | | |
| No | 639 689 (95.5) | 6412 (94.7) | 633 277 (95.5) | 1 (Ref) | 1 (Ref) |
| Yes | 29 203 (4.5) | 365 (5.3) | 28 838 (4.5) | 1.20 (1.05 to 1.36) | 1.22 (1.07 to 1.38) |
| Intrapartum haemorrhage | | | | | |
| No | 622 206 (93.3) | 5559 (82.6) | 616 647 (93.4) | 1 (Ref) | 1 (Ref) |
| Yes | 46 686 (6.7) | 1218 (17.4) | 45 468 (6.6) | 2.97 (2.75 to 3.21) | 2.75 (2.54 to 2.97) |
| Antepartum haemorrhage | | | | | |
| No | 650 631 (97.2) | 6488 (96.0) | 644 143 (97.2) | 1 (Ref) | 1 (Ref) |
| Yes | 18 261 (2.8) | 289 (4.0) | 17 972 (2.8) | 1.44 (1.25 to 1.67) | 1.29 (1.11 to 1.50) |
| Obstructed labour | | | | | |
| No | 602 184 (90.1) | 5037 (73.4) | 597 147 (90.2) | 1 (Ref) | 1 (Ref) |
| Yes | 66 708 (9.9) | 1740 (26.6) | 64 968 (9.8) | 3.35 (3.13 to 3.58) | 3.45 (3.19 to 3.74) |
| Breech presentation | | | | | |
| No | 645 162 (96.2) | 6092 (89.1) | 639 070 (96.3) | 1 (Ref) | 1 (Ref) |
| Yes | 23 730 (3.8) | 685 (10.9) | 23 045 (3.7) | 3.18 (2.88 to 3.50) | 2.80 (2.51 to 3.12) |
| Abnormal fetal position | | | | | |
| No | 640 015 (95.4) | 6269 (92.5) | 633 746 (95.5) | 1 (Ref) | 1 (Ref) |
| Yes | 28 877 (4.6) | 508 (7.5) | 28 369 (4.5) | 1.72 (1.54 to 1.92) | 1.58 (1.40 to 1.77) |

Frequencies are true counts, % and OR are weighted for design effects and non-response.
*Although the subpopulation was restricted to women who had complete data on pregnancy-specific complication, the other variables adjusted for in the model had missing data, therefore sensitivity analyses were conducted.
†Each multivariable logistic regression model adjusts for socioeconomic, health seeking behaviour, and biodemographic characteristics identified to be significantly associated with Stillbirth in the previous model.

disparities in stillbirth still exist and maternal and fetal complications were found to be important preventable risk factors. Another important finding was the risk associated with chewing tobacco which demonstrates an urgent need for strong public health messages to stop chewing tobacco during pregnancy in addition to the messages to stop smoking. Improving uptake of ANC and timely identification and effective management of maternal and fetal complications could reduce preventable stillbirths. India has a large cadre of frontline healthcare workers or community health and nutrition workers called 'ASHAs' and 'Anganwadi workers'. They could play an important role in timely identification of danger signs through frequent interactions with pregnant women who are at a higher risk of stillbirth. In addition, ASHAs and Anganwadi workers are ideally placed to facilitate information, education and communication (IEC) programmes to

specifically target stigma around reporting of stillbirth. Eradicating poverty and promotion of female education are part of the global developmental agenda espoused by the Sustainable Development Goals, for which India is a signatory. This study showed that progress towards these goals could accelerate progress in preventing stillbirths.

Despite limitations, as highlighted by Lawn et al,[32] household surveys remain the primary source of stillbirth data for LMICs with more than 75% of the global burden of stillbirths. AHS is one of the largest household surveys in the world conducted by the Office of the Registrar General & Census Commissioner of India and is therefore an important data source for preliminary and baseline studies for generating hypothesis for further in-depth research. Studies are needed to identify risk factors separately for antepartum and intrapartum stillbirths in India. The association between the maternal and

fetal complications and stillbirth suggests that although at present pregnant women in India are incentivised to seek ANC and give birth in health institutions, the quality of pregnancy care needs further investigation. Currently, neither are all stillbirths recorded nor are local reviews of stillbirths conducted in the country. Targeting at-risk population groups, recording all stillbirths and conducting local reviews would be important to reduce the high burden of stillbirths in India.

**Author affiliations**
[1]Nuffield Department of Population Health, University of Oxford, Oxford, UK
[2]National Perinatal Epidemiology Unit, Nuffield Department of Population Health, University of Oxford, Oxford, UK
[3]Department of Obstetrics and Gynaecology, FAA Medical College and Hospital, Barpeta, Assam, India
[4]Department of Obstetrics and Gynaecology, Institute of Medical Sciences, Banaras Hindu University, Ajagara, Banaras Hindu University, Varanasi, Uttar Pradesh, India
[5]Srimanta Sankaradeva University of Health Sciences, Assam, Narkashur Hilltop, Christian Basti Bhangagarh, Guwahati, Assam, India

**Contributors** NA reviewed the literature, conducted the analysis and wrote the first draft of the paper; CC helped with the data analysis, and edited the paper; SSC helped in acquiring and interpreting the data, and edited the paper; AR and UCS edited the paper; MK contributed to the interpretation and discussion of the results, and edited the paper; MN developed the concept for the study, supervised the data analysis, interpretation and discussion of the results and edited the paper.

**Funding** The study was funded by a Medical Research Council Career Development Award to Manisha Nair (Grant Ref: MR/P022030/1).

**Disclaimer** The funder had no role in the study design, data analysis, data interpretation or writing of the manuscript. All authors, had full access to all of the data in the study and can take responsibility for the integrity of the data and the accuracy of the data analysis.

**Competing interests** None declared.

**Patient consent** Not requried.

**Provenance and peer review** Not commissioned; externally peer reviewed.

**Data sharing statement** The anonymised data are freely available through the Indian Government's Data Sharing Portal.

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
