## [Reviewer comments · BMJ Open]

This paper was submitted to a another journal from BMJ but declined for publication following peer review. The authors addressed the reviewers' comments and submitted the revised paper to BMJ Open. The paper was subsequently accepted for publication at BMJ Open.

(This paper received three reviews from its previous journal but only two reviewers agreed to published their review.)

ARTICLE DETAILS

TITLE (PROVISIONAL)	Stillbirth among women in nine states in India: Rate and risk factors in study of 886,505 women from the Annual Health Survey
AUTHORS	Altijani, Noon; Carson, Claire; Choudhury, Saswati; Rani, Anjali; Sarma, Umesh C; Knight, Marian; Nair, Manisha

VERSION 1 – REVIEW

REVIEWER	Manas Pratim Roy Dte. GHS, India
REVIEW RETURNED	11-Mar-2018

GENERAL COMMENTS	“Findings are therefore generalisable to India.” - These nine states are poor performing ones. With no sample from rest of the states, the result cannot be extrapolated for the entire country. Marked variation between states is evident from the study. The role that could be played by front-line health workers in reducing still birth is absent in the message conveyed by the article. Stigma is a known reason for under-reporting. There is no recommendation for reducing it.
---

REVIEWER	Rakhi Dandona Public Health Foundation of India India
REVIEW RETURNED	15-Mar-2018

GENERAL COMMENTS	Not much is known about epidemiology and determinants of stillbirths in India. However, there are major limitations of the analysis presented which have implications on the interpretation of the data/results. These are detailed below. 1. The biggest issue with this data is the stillbirth identification based on which the entire analysis is presented. AHS questionnaire simply documents a birth as stillbirth without confirming for any sign of life. The misreporting of neonates who die immediately post birth as stillbirth is well documented (over-reporting of stillbirths). And the vice versa is true also given the stigma associated with stillbirth (under-reporting of stillbirths). Furthermore, stillbirth is defined based on the gestation period at the time of delivery. However, there is no documentation of gestation
--

	period for the births in AHS. Therefore, it seems incorrect to estimate stillbirths based on AHS data. Recent population-level data on stillbirths is available from the state of Bihar for comparison (PLOS Medicine 2017; https://doi.org/10.1371/journal.pmed.1002363), which is also in the AHS survey. It would be useful to assess the differences in the estimates and the reasons for the same between the two data sources. 2. There are no details of how stillbirth was ascertained in the survey in the methodology section. 3. AHS documents births over three-year period. It is not clear from the analysis presented if the stillbirth estimates presented in the paper are based on the entire three-year data or were annualised or were estimated based on a certain calendar year of birth. The estimates will differ based on the data utilised. 4. AHS has a very large sample size. And with such a large sample size, even a miniscule difference can become statistically significant even though the difference is so small that it has no practical consequence. Therefore, with large samples, it's critical to evaluate the practical implications of a statistically significant difference. This is evident from the regression results. The authors need to review their results within this limitation and discuss the findings that are of relevance. 5. A variety of logistic regression models are presented in the paper. The most appropriate seems the one in Table S1 which adjusts for all available variables in one model. It would be better to present this model as the main model, and discuss the findings based on this model. 6. The risk of stillbirth with chewing tobacco is being over-interpreted by the authors. The combined model as risk of borderline statistical significance. 7. Discussion is mainly about the risk factors identified and not much is written about the accuracy or reliability of these estimates. No comparisons are made with the other available data including the Sample Registration System. Because of the major limitation of definition of stillbirth per se, such a discussion is important. 8. Discussion, page 18, paragraph 1 – AHS data cannot be generalised for India even though these states account for 50% of the country's population. The states covered by AHS are poorer with higher child mortality and lower social determinants of health, which is why these states are "high-focus states" for the Indian government. 9. Introduction, paragraph 2 – With INAP, it is the first time that stillbirth has featured on the Indian government's agenda. The government efforts being referred to here were for neonatal, infant and under-5 mortality. This should not be interpreted as efforts to reduce stillbirths.
--	---

REVIEWER	Hannah Blencowe London School of Hygiene and Tropical Medicine
REVIEW RETURNED	15-Mar-2018

GENERAL COMMENTS	The authors are to be congratulated on a very clearly presented and thoughtful analysis of stillbirths in the Indian Annual Health Survey. The paper is well written, the methods appropriate and clearly presented, and sensitivity analyses undertaken and presented in the supplementary material. The discussion and conclusions are balance and based upon the findings of the study. I have a couple of minor suggestions for the manuscript for the authors to consider. The Authors note that the stillbirth rate calculated from the survey is approximately half the estimated stillbirth rate for India in recent WHO estimates, the later are also only estimates and subject to wide uncertainty bounds. The authors state that this may be due to under-reporting of stillbirths, and it could be helpful for the reader to cite research that has shown under-reporting of stillbirths in household surveys. The authors could use standard terms to describe errors in survey estimates to elaborate the potential causes of under-reporting eg omission (women not mentioning the pregnancy resulting in a stillbirth eg for fear of the implications of discussing this) or displacement (the pregnancy is captured in the survey but misclassified as a miscarriage (pregnancy loss before 7 months) or as a neonatal death. Some discussion good be added as to whether this under-reporting is likely to be differential – as this may also lead to bias in the results. Of note from Table 2 – the reported stillbirth rates in Madhya Pradesh and Chhattisgarh are similar to those reported from some high income settings. (4.2 and 5 per 1000) P5 line 8-9, discusses a ‘sub-sample’ of the population – what subsample is this – is this subsample biased at all to exclude stillbirths, from the later part of the results it seems that the subsample are those with complete data on complications during pregnancy – and it is possible that these information may be more missing for births ending in a stillbirth when compared to a livebirth. P7 line 19 – 20 – please provide further explanation as to why the ‘missing indicator’ model was chosen over the other approaches to examine missingness. P8 lines 27 – 31 – description of Schedule caste and tribe is provided in the discussion, but it may be clearer for the reader to provide some explanation at first mention P8 lines 55 – 56 – I am not following what this refers to, as does not appear to be consistent with the results shown in Table 4 P10 – table 3 current smoking unadjusted OR column has text ‘add this’ rather than results presented P12 – table 4 – what does the * in the box for mode of delivery under aOR column refer to? P16 – paragraph beginning on line 47. This paragraph refers to factors affecting the increased stillbirth odds in religious minority groups and seems to suggest that these are all through healthcare seeking behaviours – it is possible that other factors associated with increased risk of stillbirth in other studies, but not adjusted for directly in this study are important eg nutritional, exposure to environmental toxin through occupation or household (eg indoor air pollution) or exposure to domestic violence P17 line 13 – 14% higher odds – compared to not chewing tobacco or to smoking tobacco? P17 line 14 – can refs be provided for ‘other studies’? P17 paragraph beginning line 22 – the need for quality of care for effective coverage and to prevent stillbirths could be highlighted, these current measures only capture contact points with healthcare system and not content or quality of care provided P18 line 13 – the previous lines state that the survey was a
--

	representative sample from 9 states, but were the states representative of India? This would be important in interfering generalisability of these findings to the whole of India P18 line 26 – it is not clear to me why anaemia is a dummy variable which would minimise the risk of recall bias.
--	--

REVIEWER	Manmeet Kaur School of Public Health, Post-Graduate Institute of Medical Education and Research, Chandigarh, India
REVIEW RETURNED	26-Mar-2018

GENERAL COMMENTS	There are two major problems. One research design. Explain how a survey data is cross sectional design. The second major problem is of ethics. There is no mention of permissions for using data. How the raw data was secured. It also relates to the question, whether appropriate ethical measures were taken while carrying out survey. There is need to explain all of above.
--

VERSION 1 – AUTHOR RESPONSE

Reviewer: 1

Reviewer Name: Manas Pratim Roy

Institution and Country: Dte. GHS, India Please state any competing interests or state 'None declared': None declared

"Findings are therefore generalisable to India." - These nine states are poor performing ones. With no sample from rest of the states, the result cannot be extrapolated for the entire country. Marked variation between states is evident from the study.

Response: We thank the reviewer for the comments and agree that the findings cannot be generalizable for the country as a whole. We have therefore edited the discussions section mentioning that the findings are generalizable for high burden states, but may not be generalizable for the rest of India.

The role that could be played by front-line health workers in reducing still birth is absent in the message conveyed by the article.

Response: We thank the reviewer for the suggestion. As advised, we have included the following paragraph about the role of front-line healthcare workers in reducing stillbirth in India in the conclusion section:

"India has a large cadre of frontline healthcare workers or community health and nutrition workers called 'ASHAs' and 'Anganwadi workers'. They could play an important role in timely identification of danger signs through frequent interactions with pregnant women who are at a higher risk of stillbirth. In addition, ASHAs and Anganwadi workers are ideally placed to facilitate Information, Education and Communication (IEC) programmes to specifically target stigma around reporting of stillbirth".

Stigma is a known reason for under-reporting. There is no recommendation for reducing it.

Response: We agree with the reviewer and have reported the issue related to stigma being a reason for under reporting in our paper. As suggested, we have now added a recommendation for reducing stigma in the conclusion section (paragraph included in the above response).

Reviewer: 2

Reviewer Name: Rakhi Dandona

Institution and Country: Public Health Foundation of India, India Please state any competing interests or state 'None declared': None declared

Not much is known about epidemiology and determinants of stillbirths in India. However, there are major limitations of the analysis presented which have implications on the interpretation of the data/results. These are detailed below.

1. The biggest issue with this data is the stillbirth identification based on which the entire analysis is presented.

AHS questionnaire simply documents a birth as stillbirth without confirming for any sign of life. The misreporting of neonates who die immediately post birth as stillbirth is well documented (over-reporting of stillbirths). And the vice versa is true also given the stigma associated with stillbirth (under-reporting of stillbirths).

Response: We thank the reviewer for the comments. We have acknowledged the limitation of under-reporting in the paper and have added possible bias due to over-reporting in the revised draft as highlighted by the reviewer. Christou et al, 2017¹ have discussed the problem of under-reporting of stillbirth in household surveys such as the Demographic Health Surveys (DHS) across all countries globally. However, as highlighted by Lawn et al, 2010², household surveys remain the primary source of stillbirth data for low-and-middle income countries with more than 75% of the global burden of stillbirths. AHS is one of the largest household surveys in the world conducted by the Office of the Registrar General & Census Commissioner of India and is therefore an important data source for preliminary and baseline studies for generating hypothesis for further in-depth research.

Furthermore, stillbirth is defined based on the gestation period at the time of delivery. However, there is no documentation of gestation period for the births in AHS. Therefore, it seems incorrect to estimate stillbirths based on AHS data.

Response: We thank the reviewer for raising this point. It is correct that gestational age at delivery is not reported in the AHS. The outcome of our study (stillbirth) is generated from the question on outcome of last pregnancy in which the women report having either a live birth, stillbirth or abortion. If a woman reported undergoing an abortion, the gestational month of abortion was recorded. As mentioned in the above response, household survey data are not the ideal source of data for stillbirth, but at present they are the only source of data for a majority of the countries. In the study conducted by the reviewer, probing questions were used in the household survey to improve reliability of reporting of stillbirth by women and the information was further verified using verbal autopsy. This has been suggested as a reliable method by Lawn et al², but currently it is not being used in DHS and other large household surveys due to cost implications.

Recent population-level data on stillbirths is available from the state of Bihar for comparison (PLOS Medicine 2017; <https://doi.org/10.1371/journal.pmed.1002363>), which is also in the AHS survey. It would be useful to assess the differences in the estimates and the reasons for the same between the two data sources.

Response: We thank the reviewer for the suggestion. We have cited this study conducted by the reviewer and her colleagues in our revised draft. As suggested, we have now compared the AHS data from the state of Bihar with the household survey data from the reviewer's study. The rate estimated for Bihar from the AHS (11.3 per 1000 births; 95% CI = 10.9-11.8 reported in Table 2 in the paper) is about half of that reported in the reviewer's study (21.2 per 1,000 births; 95% CI 19.7 to 22.6) which is consistent with our discussion point on page 16 (lines 28-32) "The overall estimated stillbirth rate in the study population from the nine states in India was approximately half that of the World Health Organisation (WHO) estimated rate of 22 per 1000 total births." However, as pointed out by the third reviewer, we ought to acknowledge that the rates of stillbirth from the WHO, from our study and from the second reviewer's study are all estimates, and it is difficult to confirm which estimate is the most accurate.

Importantly, we found that the findings of our study in relation to the risk factors for stillbirth is consistent with the findings of the reviewer's paper which improves the reliability of our study results. We have included this comparison in our revised draft and we thank the review again for suggesting this comparison.

2. There are no details of how stillbirth was ascertained in the survey in the methodology section.

Response: Since this is a secondary data analysis, we do not have information about how stillbirth was ascertained during the survey. However, as mentioned in our response above, we have explained how we generated the outcome variable from the available dataset. As suggested, we would be happy to discuss this as a limitation in the revised draft.

3. AHS documents births over three-year period. It is not clear from the analysis presented if the stillbirth estimates presented in the paper are based on the entire three-year data or were annualised or were estimated based on a certain calendar year of birth. The estimates will differ based on the data utilised.

Response: As explained above, this data is from the Women schedule (Section-1) that was implemented during the baseline round of AHS. Ever married women in the age group 15-49 years were asked about the outcome of their last pregnancy during the reference period 01-January-2007 to 31-December-2009. The numerator 'stillbirth' and denominator 'total birth = stillbirth + livebirth' is from the same reference period of the survey data.

The AHS had first and second updation rounds after the baseline survey during which a sample of the baseline households were re-visited. As mentioned in the AHS report (available from http://www.censusindia.gov.in/vital_statistics/AHS/AHS_report_part1.pdf), only the household questionnaire was implemented in the revisited sample in all three rounds to update the data on background characteristics of the listed members of the household (example Sex, Relationship to Head, Date of Birth, Age, Religion, Social Group, Marital Status, Date at first Marriage, Education and Occupation/Activity Status).

4. AHS has a very large sample size. And with such a large sample size, even a miniscule difference can become statistically significant even though the difference is so small that it has no practical consequence. Therefore, with large samples, it's critical to evaluate the practical implications of a statistically significant difference. This is evident from the regression results. The authors need to review their results within this limitation and discuss the findings that are of relevance.

Response: We agree with the reviewer that in large datasets clinical relevance is more important than statistical significance. However the results of our regression analysis show that all the risk factors were statistically significant with a reasonably good effect size that would be considered as clinically relevant. The smallest statistically significant adjusted odds ratio that we detected was 1.11 which is a more than 10% increase in risk and is therefore clinically significant.

5. A variety of logistic regression models are presented in the paper. The most appropriate seems the one in Table S1 which adjusts for all available variables in one model. It would be better to present this model as the main model, and discuss the findings based on this model.

Response: We thank the reviewer for the comment. Table S1 presents the result of all the different methods used for the logistic regression analysis to account for the missing data. This table also includes the results of our main model shown in Table 3. We are happy to present the information on missing data analysis in the main manuscript, but we are constrained by the number of tables that we can include in the main manuscript.

6. The risk of stillbirth with chewing tobacco is being over-interpreted by the authors. The combined model as risk of borderline statistical significance.

Response: We thank the reviewer for the comment. We believe that the reviewer is referring to the complete case analysis. The effect size of the adjusted odds ratios are not materially different between our main model and the complete case analysis. However, 95% CI in the complete case analysis is wider and the p value is not statistically significant at 5% because of a smaller sample size in the complete case analysis compared with the main model which includes all data including observations with missing information.

7. Discussion is mainly about the risk factors identified and not much is written about the accuracy or reliability of these estimates. No comparisons are made with the other available data including the Sample Registration System. Because of the major limitation of definition of stillbirth per se, such a discussion is important.

Response: We thank the reviewer for the comment and agree that it is important to discuss the reliability of the estimates. We have now included a discussion comparing the estimates for the state of Bihar from the AHS data and the reviewer's paper published in Plos medicine again highlighting the under reporting of stillbirth. As far as we know, rate of stillbirth is not available from the Sample Registration System.

8. Discussion, page 18, paragraph 1 – AHS data cannot be generalised for India even though these states account for 50% of the country's population. The states covered by AHS are poorer with higher child mortality and lower social determinants of health, which is why these states are “high-focus states” for the Indian government.

Response: We thank the reviewer for the comment, and have addressed this in our response to the comments from the first reviewer.

9. Introduction, paragraph 2 – With INAP, it is the first time that stillbirth has featured on the Indian government's agenda. The government efforts being referred to here were for neonatal, infant and under-5 mortality. This should not be interpreted as efforts to reduce stillbirths.

Response: We have revised the introduction paragraph as suggested by the reviewer.

Reviewer: 3

Reviewer Name: Hannah Blencowe

Institution and Country: London School of Hygiene and Tropical Medicine Please state any competing interests or state 'None declared': None

The authors are to be congratulated on a very clearly presented and thoughtful analysis of stillbirths in the Indian Annual Health Survey. The paper is well written, the methods appropriate and clearly presented, and sensitivity analyses undertaken and presented in the supplementary material. The discussion and conclusions are balance and based upon the findings of the study.

Response: We thank the reviewer for the comments highlighting the robustness of our analysis and the usefulness of secondary data analysis using an important data source.

I have a couple of minor suggestions for the manuscript for the authors to consider. The Authors note that the stillbirth rate calculated from the survey is approximately half the estimated stillbirth rate for India in recent WHO estimates, the later are also only estimates and subject to wide uncertainty bounds. The authors state that this may be due to under-reporting of stillbirths, and it could be helpful for the reader to cite research that has shown under-reporting of stillbirths in household surveys.

Response: We thank the reviewer for this helpful comment. As mentioned in our response to the second reviewer, we agree that it is difficult to ascertain which rate is the actual representation of the population level problem of stillbirth in India, since all are estimates. As suggested by the second reviewer, we have now compared the rates from our analysis with the findings of another study in one state in India that used both household survey and verbal autopsy to estimate the rate of stillbirth in the population. In addition, as suggested we have included Christou et al, 2017¹ paper that discusses the problem of under-reporting of stillbirth in household surveys such as the Demographic Health Surveys (DHS) across all countries globally.

The authors could use standard terms to describe errors in survey estimates to elaborate the potential causes of under-reporting eg omission (women not mentioning the pregnancy resulting in a stillbirth eg for fear of the implications of discussing this) or displacement (the pregnancy is captured in the survey but misclassified as a miscarriage (pregnancy loss before 7 months) or as a neonatal death. Some discussion good be added as to whether this under-reporting is likely to be differential – as this may also lead to bias in the results. Of note from Table 2 – the reported stillbirth rates in Madhya Pradesh and Chhattisgarh are similar to those reported from some high income settings. (4.2 and 5 per 1000)

Response: We thank the reviewer for these comments. These are important points and we have therefore included the following paragraph in the discussion section about the likelihood of differential under-reporting and misclassification:

“A study by Christou et al, 2017¹ showed that stillbirths are likely to be under-reported in routine household surveys, but this was more likely to be due a lack of rigorous ascertainment of pregnancy outcomes rather than deliberate non-reporting by women due to any reason. We did not find any evidence from published literature suggesting under-reporting of stillbirth to vary by risk factors or specific population groups, thus we excluded the possibility of differential under-reporting. However, we cannot exclude the possibility of misclassification of stillbirth as miscarriage/ abortion or neonatal deaths as stillbirth. A small proportion of the stillbirths were reported as pregnancy loss/ abortion after 7 months of gestation in the dataset. We reclassified these as stillbirth in our analysis.”

P5 line 8-9, discusses a ‘sub-sample’ of the population – what subsample is this – is this subsample biased at all to exclude stillbirths, from the later part of the results it seems that the subsample are those with complete data on complications during pregnancy – and it is possible that these information may be more missing for births ending in a stillbirth when compared to a livebirth.

Response: Information on pregnancy complications was missing for >25% of the original sample. Due to this reason we limited the specific analysis of pregnancy complications to a subsample with complete information. We compared the proportion of stillbirth in the subsample with the excluded group and the total sample which showed that in all three groups the proportion of stillbirth was about 1% and livebirth 99%. This suggests that the subsample for the specific pregnancy complication analysis was not a biased sample. We have included this in explanation in the methods section of the revised draft.

P7 line 19 – 20 – please provide further explanation as to why the ‘missing indicator’ model was chosen over the other approaches to examine missingness.

Response: We thank the reviewer for this question. The main reason for using the missing indicator model as the main model was to have as much power in the analysis as possible. The main advantages of this method is that we were able to retain all observations including those with missing information as well as examine the association between the missing category for each risk factor and the outcome variable since the missing category might be informative in its own right.

P8 lines 27 – 31 – description of Schedule caste and tribe is provided in the discussion, but it may be clearer for the reader to provide some explanation at first mention

Response: We have described Schedule Caste and Tribe social groups in Table 1 in the methods section.

P8 lines 55 – 56 – I am not following what this refers to, as does not appear to be consistent with the results shown in Table 4

Response: We apologise for the lack of clarity. We have updated this paragraph in the revised version. We have moved the following sentence from page 8 lines 55 – 56 to the paragraph reporting association between specific complications and stillbirth, since the attenuation of the association between caesarean section and stillbirth was specifically observed in relation to the obstructed labour model.

“However, the association between stillbirth and caesarean section was no longer significant at $p < 0.05$ and the aOR for assisted delivery was significantly attenuated in the model that examined the association between obstructed labour and stillbirth.”

P10 – table 3 current smoking unadjusted OR column has text ‘add this’ rather than results presented

Response: We apologise for this typographical error, and have now rectified this.

P12 – table 4 – what does the * in the box for mode of delivery under aOR column refer to?

Response: We were not able to locate this issue on the submitted version. We have checked all tables to rectify any other typographical errors.

P16 – paragraph beginning on line 47. This paragraph refers to factors affecting the increased stillbirth odds in religious minority groups and seems to suggest that these are all through healthcare seeking behaviours – it is possible that other factors associated with increased risk of stillbirth in other studies, but not adjusted for directly in this study are important eg nutritional, exposure to environmental toxin through occupation or household (eg indoor air pollution) or exposure to domestic violence

Response: We agree with the reviewer and have added the suggestions to the discussion section in the revised version.

P17 line 13 – 14% higher odds – compared to not chewing tobacco or to smoking tobacco?

Response: We thank the reviewer for pointing this out. We have clarified this sentence as below: "In the study population, chewing tobacco was 15-times more common compared with smoking tobacco and women who chewed tobacco had a 14% higher odds of stillbirth compared with women who did not"

P17 line 14 – can refs be provided for 'other studies'?

Response: As suggested by the reviewer, we have now added the references.

P17 paragraph beginning line 22 – the need for quality of care for effective coverage and to prevent stillbirths could be highlighted, these current measures only capture contact points with healthcare system and not content or quality of care provided

Response: We agree with the reviewer and have added the suggested point about the importance of quality of care, in addition to coverage.

P18 line 13 – the previous lines state that the survey was a representative sample from 9 states, but were the states representative of India? This would be important in interfering generalisability of these findings to the whole of India

Response: We thank the reviewer for the comment, and have addressed this in our response to the comments from the first reviewer.

P18 line 26 – it is not clear to me why anaemia is a dummy variable which would minimise the risk of recall bias.

Response: We thank the reviewer for the question. We suspect that women would be unlikely to report anaemia as a specific complication during pregnancy since they might be unaware of their haemoglobin concentration. However, derived variable based on questions related to the following signs and symptoms: paleness, giddiness, weakness, excessive fatigue, provided information about anaemia which was unlikely to be due to recall bias or misreporting. In Table 1 in the revised draft, we have described the original data from which the dummy variable 'anaemia' was generated.

Reviewer: 4

Reviewer Name: Manmeet Kaur

Institution and Country: School of Public Health, Post-Graduate Institute of Medical Education and Research, Chandigarh, India Please state any competing interests or state 'None declared': None declared

There are two major problems. One research design. Explain how a survey data is cross sectional design.

Response: We thank the reviewer for the comment. The survey had three successive rounds of data collection, and our study is a cross sectional analysis using data from round 1 only.

The second major problem is of ethics. There is no mention of permissions for using data. How the raw data was secured. It also relates to the question, whether appropriate ethical measures were taken while carrying out survey. There is need to explain all of above.

Response: The AHS was conducted by the Office of the Registrar General & Census Commissioner, India (http://censusindia.gov.in/vital_statistics/AHSBulletins/ahs.html) who were responsible for appropriate ethical measures. There are no ethical implications for analysing anonymised secondary data from the survey that is available in the public domain as mentioned in the manuscript: “The anonymised data is freely available through the Indian Government’s Data Sharing Portal.”

References

1. Christou A, Dibley MJ, Raynes-Greenow C. Beyond counting stillbirths to understanding their determinants in low-and middle-income countries: a systematic assessment of stillbirth data availability in household surveys. *Tropical Medicine & International Health* 2017; **22**(3): 294-311.
2. Lawn JE, Gravett MG, Nunes TM, Rubens CE, Stanton C. Global report on preterm birth and stillbirth (1 of 7): definitions, description of the burden and opportunities to improve data. *BMC Pregnancy and Childbirth* 2010; **10**(1): S1.

VERSION 2 – REVIEW

REVIEWER	Manas Pratim Roy Ministry of Health and Family Welfare, India
REVIEW RETURNED	18-May-2018

GENERAL COMMENTS	“...smoking was not found to be significantly associated with stillbirth in our study population after adjusting for other risk factors...”- For smoking, unadjusted Odd’s ratio is still 1.53 for still birth, as evident from Table 3. Even if not significant, the figures could be given in table.
--

REVIEWER	Rakhi Dandona PHFI, New Delhi, India
REVIEW RETURNED	29-May-2018

GENERAL COMMENTS	The authors have missed the point about under-reporting of stillbirths and the methodology of documenting the stillbirths in the Annual Health Survey. The point that household surveys remain the only available source for stillbirth estimation or that the AHS is a large survey is not the question here. The issue here is the validity of these estimates, and that needs to be clearly documented in the manuscript. The comments on the previous version of this manuscript are not appropriately addressed in this revision. 1. AHS is indeed a large survey, however, that by itself does not mean that the methods to ascertain stillbirth can be overlooked. The limitations of the data should be clearly mentioned for the readers. There is still considerable lack of discussion on limitation of these data. 2. The authors have reported details of data collection methods in their response to the previous review point 1. These details should be provided in the manuscript - documentation of stillbirth without confirmation of life and lack of gestation period documentation. 3. In response to previous review point 1, the authors state “as mentioned in the above response, household survey data are not the ideal source of data for stillbirth, but at present they are the only source of data for a majority of the countries..... This has been suggested as a reliable method by Lawn et al, but currently it is not
--

	being used in DHS and other large household surveys due to cost implications.” This interpretation is not fully accurate. Household surveys can provide the stillbirth data if the gestation period is documented, and of course further improvements can be made as suggested by Lawn et al. However, estimating stillbirth with no data on gestation period at all is questionable (and stillbirth is defined based on gestation period) irrespective of the survey in any part of the world. 4. In response to previous review point 1, the authors mention “however, as pointed out by the third reviewer, we ought to acknowledge that the rates of stillbirth from the WHO, from our study and from the second reviewer’s study are all estimates, and it is difficult to confirm which estimate is the most accurate”. This interpretation is not fully correct. In reality, none of the estimates may be fully accurate, however, it is important to be upfront about the limitations of each of these estimates. It is very clear that with the AHS data that there are significant limitations in the data capture, which are not discussed appropriately in this manuscript. 5. In response to previous review point 2, the authors mention that “since this is a secondary data analysis, we do not have information about how stillbirth was ascertained during the survey”. The AHS survey questionnaire is readily available in public domain to ascertain what was asked in the interview.
--	---

REVIEWER	Hannah Blencowe London School of Hygiene and Tropical Medicine
REVIEW RETURNED	17-May-2018

GENERAL COMMENTS	The authors have provided clear responses to my and the other reviewers comments. The revised manuscript is clearer and will make a useful contribution to the published literature
---

VERSION 2 – AUTHOR RESPONSE

Reviewer: 1

Reviewer Name: Manas Pratim Roy

Institution and Country: Ministry of Health and Family Welfare, India Please state any competing interests or state 'None declared': None declared

Please leave your comments for the authors below

“...smoking was not found to be significantly associated with stillbirth in our study population after adjusting for other risk factors...”- For smoking, unadjusted Odd’s ratio is still 1.53 for still birth, as evident from Table 3. Even if not significant, the figures could be given in table.

Response: We thank the reviewer for the comment. We used a set protocol to build the statistical model and decided apriori that variables that are not significant during the model building process would not be included in the final model. This does not allow us to add the adjusted odds ratio to the table. However, we have added the adjusted OR in the discussion section on page 17: “In contrast to other studies, 16 17 smoking was not found to be significantly associated with stillbirth in our study population after adjusting for other risk factors (aOR = 1.26; 95% CI =0.97 to 1.63).”

Reviewer: 2

Reviewer Name: Rakhi Dandona

Institution and Country: PHFI, New Delhi, India Please state any competing interests or state 'None declared': No competing interests

Please leave your comments for the authors below.

The authors have missed the point about under-reporting of stillbirths and the methodology of documenting the stillbirths in the Annual Health Survey. The point that household surveys remain the only available source for stillbirth estimation or that the AHS is a large survey is not the question here. The issue here is the validity of these estimates, and that needs to be clearly documented in the manuscript. The comments on the previous version of this manuscript are not appropriately addressed in this revision.

1. AHS is indeed a large survey, however, that by itself does not mean that the methods to ascertain stillbirth can be overlooked. The limitations of the data should be clearly mentioned for the readers. There is still considerable lack of discussion on limitation of these data.

Response: We agree with the reviewer that methods to ascertain stillbirths should not be overlooked and as was suggested we expanded the limitations section in the first revision to clearly state that "stillbirths may be under-reported or misclassified in the AHS from which our data were drawn" (Page-18). The following sentence was also included in the limitations section "we cannot exclude the possibility of misclassification of stillbirth as miscarriage/ abortion or neonatal deaths as stillbirth." (Page-18)

2. The authors have reported details of data collection methods in their response to the previous review point 1. These details should be provided in the manuscript - documentation of stillbirth without confirmation of life and lack of gestation period documentation.

Response: As suggested by the reviewer, we have added the following in the methods section – "Ever married women in the age group 15-49 years were asked about the outcome of their last pregnancy during the reference period 01-January-2007 to 31-December-2009, which was reported as either livebirth, stillbirth or abortion. Information on gestational age at stillbirth or type of stillbirth (antepartum or intrapartum) was not available."

We have further added the following paragraph in the limitations section –

"We acknowledge that household surveys are not the ideal source of data for stillbirth, but at present they are the only source of data for a majority of the countries. The reliability of the reporting of stillbirth in household surveys could be improved by including information on gestational age at stillbirth, probing questions as was done in a study by Dandona et al conducted in one state in India⁹ or by using verbal autopsy,³² but currently it is not being used in large household surveys due to cost implications." (Pages 18-19)

3. In response to previous review point 1, the authors state "as mentioned in the above response, household survey data are not the ideal source of data for stillbirth, but at present they are the only source of data for a majority of the countries..... This has been suggested as a reliable method by Lawn et al, but currently it is not being used in DHS and other large household surveys due to cost implications." This interpretation is not fully accurate. Household surveys can provide the stillbirth data if the gestation period is documented, and of course further improvements can be made as suggested by Lawn et al. However, estimating stillbirth with no data on gestation period at all is questionable (and stillbirth is defined based on gestation period) irrespective of the survey in any part of the world.

Response: Acknowledging the limitations, in the conclusion section we have already stated that these data could be useful for hypothesis generating and such work should be followed by further in-depth

research. Reviewer's point related to improving reporting by including information on gestational age at stillbirth has been added to the limitations section (please see our response above).

4. In response to previous review point 1, the authors mention "however, as pointed out by the third reviewer, we ought to acknowledge that the rates of stillbirth from the WHO, from our study and from the second reviewer's study are all estimates, and it is difficult to confirm which estimate is the most accurate". This interpretation is not fully correct. In reality, none of the estimates may be fully accurate, however, it is important to be upfront about the limitations of each of these estimates. It is very clear that with the AHS data that there are significant limitations in the data capture, which are not discussed appropriately in this manuscript.

Response: We would like to draw attention of the reviewer to the limitations section (Pages 18 and 19), which we expanded substantially based on comments from all reviewers clearly stating the potential issues related to under-reporting and misclassification. We have made further changes, within the scope of the study, to this section in the attached draft as suggested by the reviewer.

5. In response to previous review point 2, the authors mention that "since this is a secondary data analysis, we do not have information about how stillbirth was ascertained during the survey". The AHS survey questionnaire is readily available in public domain to ascertain what was asked in the interview.

Response: We have not been able to find the questionnaire on public domain, and have therefore transparently reported this as a limitation.

Reviewer: 3

Reviewer Name: Hannah Blencowe

Institution and Country: London School of Hygiene and Tropical Medicine Please state any competing interests or state 'None declared': None declared

Please leave your comments for the authors below

The authors have provided clear responses to my and the other reviewers comments. The revised manuscript is clearer and will make a useful contribution to the published literature.

Response: We thank the review for the constructive feedback which was helpful in revising the paper.